# Effects of smoke flavoring using different wood chips and barbecuing on the formation of polycyclic aromatic hydrocarbons and heterocyclic aromatic amines in salmon fillets

Emel Oz [ORCID] *

Department of Food Engineering, Faculty of Agriculture, Ataturk University, Erzurum, Turkey

* emel.oz@atauni.edu.tr

**Citation:** Oz E (2020) Effects of smoke flavoring using different wood chips and barbecuing on the formation of polycyclic aromatic hydrocarbons and heterocyclic aromatic amines in salmon fillets. PLoS ONE 15(1): e0227508. https://doi.org/10.1371/journal.pone.0227508

**Data Availability Statement:** All relevant data are within the paper.

**Funding:** The author(s) received no specific funding for this work.

## Abstract

Herein, the concentrations of food toxicants, polycyclic aromatic hydrocarbons (PAHs) and heterocyclic aromatic amines (HAAs), in salmon fillets smoke flavored with different smoking wood chips (oak, apple, bourbon soaked oak, cherry and hickory) and barbecuing were determined. Benzo[a]anthracene (up to 0.24 ng/g) and chrysene (0.22 ng/g) were determined in the raw salmon fillets. While ∑PAH8 (benzo[a]anthracene, chrysene, benzo[b]fluoranthene, benzo[k]fluoranthene, benzo[a]pyrene, dibenzo[a,h]anthracene, benzo[g,h,i]perylene, indeno[1,2,3-cd]pyrene) in the raw samples ranged between 0.44 and 0.46 ng/g, smoke flavoring increased the amount of ∑PAH8 and the amount varied between 0.47 and 0.73 ng/g. Salmon smoked flavored with bourbon soaked oak, cherry and hickory wood chips and barbecued showed significantly ($P$ <0.05) lower contents of ∑PAH4 (benzo[a]anthracene, chrysene, benzo[b]fluoranthene, benzo[a]pyrene) and ∑PAH8 compared to non-smoke flavored barbecued samples. Additionally, smoke flavoring with apple, bourbon soaked oak, and cherry wood chips significantly ($P$ <0.05) reduced the total HAA contents in barbecued salmon. A remarkable result was that the bourbon-soaked oak and cherry wood chips had inhibitory effects on both PAH and HAA contents. In sum, barbecued non-smoke flavored and smoke flavored salmon with different wood chips could be considered safe from the perspective of the detected amounts of PAHs and HAAs in salmon fillets.

## Introduction

Fish provides a growing number of nutrients, such as protein, omega-3 fatty acids, eicosapentaenoic acid (EPA), docosahexaenoic acid (DHA), minerals (phosphorus, iodine, iron and fluorine), and vitamins (A and D). Studies have found that fish intake could reduce the risk of coronary heart disease, lowers the incidence of diabetes, and plays an important role in the nervous system function, due to omega-3 polyunsaturated fatty acids (PUFAs) content which is very rich in the flesh of Atlantic salmon (*Salmo salar*) [1,2].

**Competing interests:** The authors have declared that no competing interests exist.

Smoking is one of the oldest food preservation methods and still in use today in fish processing [3,4]. In addition to the antimicrobial and antioxidant effects of smoking, certain organoleptic characteristics (taste, color and aroma) and texture can be achieved. Wood type, temperature, and smoking duration are among the factors that could influence the quality of smoked fish. It has to be noted that wood smoke contains some carcinogenic compounds, such as polycyclic aromatic hydrocarbons (PAHs) [3].

PAHs are a large group of persistent organic pollutants consisting of two or more fused aromatic rings [5]. Among the hundreds of PAHs, benzo[a]pyrene (BaP) classified as a group 1 (carcinogenic to humans) and 16 others assigned either to group 2A (probably carcinogenic) or group 2B (possible carcinogen) are generated during the smoking processes [6]. Reported PAHs concentrations in smoked fishes are highly variable, due to the differences in the smoking process, including wood type, smoke generation conditions, temperature of pyrolysis, and smoking duration [7]. Woodchip type (varied on the basis of chemical composition and species) has an impact on the PAH content of the smoked products. On the other hand, PAHs can also be emitted during thermal treatment, such as barbecuing and grilling [8].

Fatty fish (such as salmon) are ideal for barbecuing, the most traditional and popular way of cooking in Turkey and other countries. PAHs are primarily generated as a result of direct pyrolysis of meat fats with flame. Alternatively, fat dripping onto the flame or hot coals can produce PAH's, which are then come back up in the smoke to the meat. The formation of PAHs by barbecuing is a function of both fat content of the meat and distance from the heat source [9]. During barbecuing, some hazardous compounds known as heterocyclic aromatic amines (HAA) are also formed [10]. HAAs, mutagenic and/or carcinogenic compounds, were firstly identified in grilled meat and fish in 1977 [11]. So far, the levels of more than 25 HAA compounds have been determined in heat-treated foods [12].

The impacts of smoking parameters on the formation of PAHs in some fish fillets were reported elsewhere [13–15]. However, these studies were generally focused on smoking time and temperature. Therefore, there is a general lack of knowledge concerning the influence of wood type on the emission of PAHs. Furthermore, the effects of smoke flavoring and barbecuing on the levels of PAHs and HAAs in salmon fillets were not yet addressed in the literature. Determination of HAA and PAH contents in salmon fillets smoke flavored with different wood chips is very crucial to have a database and evaluate the relationship between dietary pattern and health outcomes. Thence, the present study was aimed to investigate the effects of smoke flavoring with various types of wood chips (oak, apple, bourbon soaked oak, cherry and hickory) and barbecuing on the levels of PAHs and HAAs in salmon fillets.

## Materials and methods

### Fish fillets preparation

Salmon fishes (n = 14), farmed in Black Sea, were procured from a local fish store (Erzurum, Turkey). Fillets (used in the present study) were prepared from frozen samples and defrosted in the laboratory.

### Chemicals

Chemicals and solvents were of high-performance liquid chromatography (HPLC) and/or analytical grade. All solutions (except HPLC-grade solvents) were passed through a 0.45μm filter (Milex®, Massachusetts, USA). Standard mixture of PAH was purchased from Supelco (Bellefonte, PA, USA), while HAA mixture standards were acquired from Toronto Research Chemicals (Downsview, Ontario, Canada).

## Smoke flavoring of salmon fillets

A smoking gun (Polyscience, the smoking gun PRO, ÖRKA, Turkey) was used for smoke flavoring. Total 28 salmon fillets obtained from 14 salmon fishes were divided into seven groups; two of them were used as the control. The first group (C-I, n = 4) was not smoke flavored and immediately barbecued (without storage). Second group (C-II, n = 4) was not smoke flavored, placed in a special zipper bag (ÖRKA, Turkey) and barbecued after storage at 24˚C for 3 h. Other five groups of salmon fillets (n = 4 salmon fillets for each group) were also placed in special zipper bags (ÖRKA, Turkey) and smoke flavored over five different wood chips, including oak (S-I), apple (S-II), bourbon soaked oak (S-III), cherry (S-IV), and hickory (S-V) (Elektrola, ÖRKA, Turkey) at 24˚C (cold smoke flavoring) for 5 min. All groups of smoked fillets (except C-I group) were stored at 24˚C for 3 h to accumulate smoke on the surface. After storage, samples were removed from zipper bags and were immediately barbecued.

## Barbecuing of salmon fillets

Salmon fillets are grilled using charcoal barbecue without the addition of frying fat/oil, salt, spice, and food additives. The grill surface temperature was approximately 200˚C (Testo 926, Lenzkirch, Germany). All samples were turned over one-half the time necessary for cooking (8 min). Two replicate experiments were conducted with 2 fillets per each replicate/condition. After barbecuing, samples were cooled at room temperature and minced using a kitchen blender (Tefal, Istanbul, Turkey).

## Analysis of polycyclic aromatic hydrocarbon in salmon fillet

The PAHs content of the samples were determined according to Farhadian et al. [16], with minor modifications. Briefly, the samples and cold NaOH solution (1 M) were homogenized and mixed with Extrelut refill material. Then, the mixture was transferred to Extrelut column and the PAH fraction was eluted with dichloromethane. The elutes were evaporated to dryness and the residues were dissolved in *n*-hexane. Again, the mixture was eluted from the column using *n*-hexane and dichloromethane and the elute was evaporated till dryness and the residue was reconstituted in acetonitrile. HPLC (Thermo Ultimate 3000; Thermo Scientific) with fluorescence detector (FLD-3000) was used for determination of PAH content in fillet samples. Separation was performed using a Hypersil™ Green PAH LC column (150 mm × 2.1 mm, 3 μm particle size, Hichrom, Reading, UK). A mixture of deionized water/acetonitrile (60:40, *v/v*) was used as a mobile phase.

## Determination of heterocyclic aromatic amines in salmon fillet

The HAA content was determined according to the method described by Messner and Murkovic [17], with minor modifications. Briefly, the barbecued samples and NaOH (1 M) were homogenized and mixed with Extrelut NT packing material (Merck, Darmstadt, Germany). During the solid-phase extraction, ethyl acetate was used for extraction. After washing with HCl and MeOH, the analytes including HAAs were eluted with MeOH-concentrated ammonia. The eluted mixtures were evaporated and the residues were dissolved in MeOH including internal standard. HPLC (Thermo Ultimate 3000, Thermo Scientific, Santa Clara, CA) with diode array detector (DAD-3000) was used for determination and Acclaim™ 120 C18, 3 μm (4.6 × 150 mm, Tosoh Bioscience GmbH, Stuttgart, Germany) column was used for separation of HAAs. A mobile phase consisting of methanol/acetonitrile/water/acetic acid (8/14/76/2, *v/v/v/v*) at pH 5.0 (adjusted with ammonium hydroxide 25%) and acetonitrile were used as solvent A and solvent B, respectively.

### Limits of detection and quantification values and recoveries of PAHs and HAAs

The limits of detection (LOD) and quantification (LOQ) of PAHs and HAAs (from freshly prepared working standard solutions) were calculated based on signal to-noise ratios of 3 and 10, respectively. Recovery rates for different PAHs and HAAs in the samples were determined by the standard addition method.

### Statistical analyses

The data obtained in the present study were subjected to analysis of variance. The experiment was set up according to a randomized design and employed in two replicates for all analyses. Duncan multi comparison test was performed to determine the statistical differences between the values, using the Statistical Package for the Social Sciences 11.5 statistical software package.

## Results and discussion

### Polycyclic aromatic hydrocarbon contents in raw salmon fillets

The limits of detection (LOD) and quantification (LOQ) as well as the recovery rates of polycyclic aromatic hydrocarbons (PAHs) analyzed in the present study are compiled in Fig 1. These values were comparable with those reported in the literature for PAHs [8].

It was found that non-smoke flavored raw salmon fillets contain low molecular weight compounds (three rings, BaA and Chry) up to 0.24 ng/g (Table 1). Environmental pollution is considered as one of the important source of PAHs in aquaculture and three or four rings-PAHs are the most representative compounds found in coastal and estuarine ecosystem [18]. Herein, BaP, DahA, BghiP and IncdP are not detected both in non-smoke flavored and smoke flavored raw salmon fillets. As well the non-smoke flavored raw salmon fillets, BbF and BkF were not detected as well in raw salmon fillets smoke flavored with bourbon soaked oak wood chips. Although, BbF and BkF were detected in other smoke flavored raw salmon fillets, however, their levels were below the limit of quantification.

BaA was detected in all raw salmon fillets at a concentration range of 0.22 to 0.41 ng/g. The highest concentration was determined in raw salmon fillets smoke flavored with oak wood chips. It was reported that commercially smoked Norwegian, Danish, and Scottish salmon contains BaA at a level of 0.5, 1.2, and 23.2 ng/g, respectively [19], and up to 1.75 µg/kg was detected in smoked salmon by Zachara et al. [20].

The concentration of Chry was ranged between 0.22 and 0.39 ng/g in all raw salmon fillets. The highest content was found in raw salmon fillets smoke flavored with cherry wood chips. Notably, Chry concentration up to 2.30 µg/kg was determined in smoked salmon by Zachara et al. [20], up to 24.47 ng/g dry weight in raw Atlantic salmon fillets and up to 15.53 ng/g dry weight in cold-smoked Atlantic salmon fillets by Visciano et al. [7].

In this study, while BbF was not detected in non-smoke flavored raw salmon fillets and smoke flavored fillets with bourbon soaked oak, cherry, and hickory wood chips, it was detected, but not quantified in raw salmon fillets smoke flavored with oak and apple wood chips. Obviously, BbF at a concentration of 0.40 µg/kg was recorded in smoked salmon by Zachara et al. [20], a level up to 8.46 and 8.07 ng/g f dry weight was found in raw Atlantic salmon fillets and cold-smoked Atlantic salmon fillets as reported by Visciano et al. [7].

We could not determine BkF in non-smoke flavored raw salmon fillets and smoke flavored with apple and bourbon soaked oak wood chips. On the other hand, BkF was detected, but not quantified in raw salmon fillets smoke flavored with oak and cherry wood chips. As recorded

| Compounds | Structure | IARC | LOD (ng/g) | LOQ (ng/g) | Recovery (%) |
|---|---|---|---|---|---|
| *Polycyclic aromatic hydrocarbons* | | | | | |
| BaA | | Group 2B | 0.027 | 0.090 | 92.57 |
| Chry | | Group 2B | 0.034 | 0.113 | 74.64 |
| BbF | | Group 2B | 0.086 | 0.288 | 87.69 |
| BkF | | Group 2B | 0.065 | 0.869 | 80.00 |
| BaP | | Group 1 | 0.069 | 0.228 | 89.23 |
| DahA | | Group 2A | 0.083 | 0.277 | 55.13 |
| BghiP | | Group 3 | 0.125 | 0.415 | 70.07 |
| IncdP | | Group 2B | 0.113 | 0.376 | 49.03 |
| *Heterocyclic aromatic amines* | | | | | |
| IQx | | | 0.004 | 0.013 | 71.35 |
| IQ | | Group 2A | 0.009 | 0.029 | 45.11 |
| MeIQx | | Group 2B | 0.024 | 0.081 | 74.67 |
| MeIQ | | Group 2B | 0.014 | 0.047 | 35.52 |
| 7,8-DiMeIQx | | | 0.005 | 0.018 | 69.20 |
| 4,8-DiMeIQx | | | 0.008 | 0.025 | 70.28 |
| PhIP | | Group 2B | 0.025 | 0.085 | 82.15 |
| AαC | | Group 2B | 0.012 | 0.039 | 98.30 |
| MeAαC | | Group 2B | 0.010 | 0.035 | 75.91 |

**Fig 1. Limits of detection (LOD), limits of quantification (LOQ), recoveries of standard curve and percent recovery of PAH and HAA standards.** IARC: International Agency for Research Cancer, Group 1: potentially carcinogenic to human, Group 2A: probably carcinogenic to humans, Group 2B: possibly carcinogenic to humans, Group 3: not classifiable as to its carcinogenicity to humans.

**Table 1. Polycyclic aromatic hydrocarbon (PAH) contents of raw and barbecued non-smoke flavored and smoke flavored salmon fillets with different smoking wood chips (ng/g).**

| Group | n | BaA | Chry | BbF | BkF | BaP | DahA | BghiP | IncdP | ΣPAH4 | ΣPAH8 |
|---|---|---|---|---|---|---|---|---|---|---|---|
| *Raw* | | | | | | | | | | | |
| C-I | 4 | 0.22±0.03 | 0.22±0.03 | nd | nd | nd | nd | nd | nd | 0.44±0.03 d | 0.44±0.03 d |
| C-II | 4 | 0.24±0.01 | 0.22±0.02 | nd | nd | nd | nd | nd | nd | 0.46±0.01 d | 0.46±0.01 d |
| S-I | 4 | 0.41±0.05 | 0.32±0.04 | nq | nq | nd | nd | nd | nd | 0.73±0.06 a | 0.73±0.06 a |
| S-II | 4 | 0.23±0.03 | 0.34±0.04 | nq | nd | nd | nd | nd | nd | 0.57±0.04 bc | 0.57±0.04 bc |
| S-III | 4 | 0.25±0.03 | 0.25±0.03 | nd | nd | nd | nd | nd | nd | 0.50±0.04 cd | 0.50±0.04 cd |
| S-IV | 4 | 0.23±0.02 | 0.39±0.05 | nd | nq | nd | nd | nd | nd | 0.62±0.04 b | 0.62±0.04 b |
| S-V | 4 | 0.22±0.07 | 0.25±0.04 | nd | nq | nd | nd | nd | nd | 0.47±0.04 d | 0.47±0.04 d |
| Sign | | | | | | | | | | ** | ** |
| *Barbecued* | | | | | | | | | | | |
| C-I | 4 | 0.59±0.17 | 0.77±0,18 | 0.53±0.13 | nq | 0.52±0.26 | nq | 0.47±0.04 | nq | 2.41±0.16 b | 2.88±0.16 b |
| C-II | 4 | 0.70±0.12 | 0.87±0.14 | 0.50±0.17 | nq | 0.49±0.24 | nq | nq | 0.40±0.22 | 2.56±0.21 b | 2.96±0.21 b |
| S-I | 4 | 0.80±0.16 | 1.08±0.13 | 0.77±0.15 | nq | 1.04±0.21 | nq | 0.66±0.05 | 0.62±0.12 | 3.69±0.18 a | 4.97±0.18 a |
| S-II | 4 | 1.06±0.14 | 1.01±0.11 | 0.82±0.18 | nq | 1.01±0.11 | nq | 0.79±0.05 | 0.49±0.10 | 3.90±0.24 a | 5.18±0.24 a |
| S-III | 4 | 0.28±0.12 | 0.33±0.14 | nq | nq | 0.23±0.08 | nd | nq | nq | 0.84±0.17 d | 0.84±0.17 d |
| S-IV | 4 | 0.34±0.09 | 0.48±0.13 | 0.41±0.16 | nq | 0.33±0.07 | nd | nq | nd | 1.56±0.14 c | 1.56±0.14 c |
| S-V | 4 | 0.27±0.08 | 0.41±0.20 | nq | nq | 0.25±0.10 | nd | nq | nd | 0.93±0.13 d | 0.93±0.13 d |
| Sign | | | | | | | | | | ** | ** |

n: number of salmon fillet; C-I: control non-smoke flavored, non-stored salmon fillets; C-II: control non-smoke flavored salmon fillets stored at 24˚C for 3 h; S-I: salmon fillets smoke flavored over oak wood chips and stored at 24˚C for 3 h; S-II: salmon fillets smoke flavored over apple wood chips and stored at 24˚C for 3 h; S-III: salmon fillets smoke flavored over bourbon soaked oak wood chips and stored at 24˚C for 3 h; S-IV: salmon fillets smoke flavored over cherry wood chips and stored at 24˚C for 3 h; S-V: salmon fillets smoke flavored over hickory wood chips and stored at 24˚C for 3 h; nd: not detected (<LOD); nq: not quantified (LOD<...<LOQ); BaA: benzo[a]anthracene; Chry: chrysene; BbF: benzo[b]fluoranthene; BkF: benzo[k]fluoranthene; BaP: benzo[a]pyrene; DahA: dibenzo[a,h]anthracene; BghiP: benzo [g,h,i]perylene, IncdP: indeno[1,2,3-cd]pyrene; ΣPAH4: BaA, Chry, BbF and BaP;ΣPAH8: BaA, Chry, BbF, BkF, BaP, DahA, BghiP and IncdP; Sign: significance
**: P <0.01; a-d: means with different letters in the same column are significantly different (P <0.05).

elsewhere [7], BkF was determined up to 1.90 ng/g dry weight in raw Atlantic salmon fillets and up to 1.46 ng/g dry weight in cold-smoked Atlantic salmon fillets.

BaP is a known carcinogen assigned by the International Agency for Research on Cancer (IARC) as a group 1 carcinogen, while the majority of other high molecular weight (HMW) PAHs (≥ five ring) are largely classified either in group 2A (probably carcinogen) or 2B (possibly carcinogen) categories [6]. Herein, BaP, DahA, BghiP, and IncdP were not estimated in any of non-smoke flavored and smoke flavored salmon fillets. In the same line, Storelli et al. [19] could not detect DahA and BghiP in commercially smoked Norwegian, Danish, and Scottish salmon fillets. On the other hand, they only found BaP in commercially smoked Scottish salmon at a level of 0.7 ng/g. Visciano et al. [7] determined BaP and BghiP up to the level of 9.88 and 8.65 ng/g dry weight in raw Atlantic salmon fillets and up to 7.71 and 10.76 ng/g dry weight in cold-smoked Atlantic salmon fillets. In this study, the BaP level was lower than the maximum limit set by the EU Commission Regulation (2 µg/kg in smoked fish, 21] for all salmon fillets smoked with different wood chips.

Results obtained in the present study showed that even the raw and non-smoke flavored salmon fillets could contain PAHs at different levels. Herein, the presence of PAHs at various levels in raw and non-smoke flavored fish could be attributed to environmental and/or sea pollution [7] or diet for farm fishes [18]. Smoke flavoring would increase the total amount of PAHs. The PAH contents of smoke flavored raw salmon fillets have all been attributed to source pollution and smoking process.

ΣPAH4, the sum of the four PAH4 (BaP, BaA, BbF, and Chry) is the most suitable marker of PAHs in food [21]. Due to the fact that the other PAHs (except for BaA and Chry) were not detected or determined in salmon fillets, herein, the ΣPAH4 was equal to ΣPAH8 in all raw salmon fillets. The results showed that ΣPAH4 varied from 0.44 to 0.73 ng/g in samples and smoke flavoring tends to elevate the amount of ΣPAH4 compared to non-smoke flavored raw salmon fillets. The lowest ΣPAH4 contents were observed in non-smoke flavored salmon fillets (C-I and C-II) and smoke flavored salmon fillets with hickory (S-V) wood chips. On the other hand, smoke flavored fillets with bourbon soaked oak (S-III) and hickory (S-V) wood chips were not statistically different from each other. The highest ΣPAH4 content was found to be in salmon fillets smoke flavored with oak wood chips. These results may be related to difference in smoke generation temperatures of wood chips. Indeed, it was reported that the smoke generation temperature could influence the PAH levels of smoked meat products [22]. The smoke generation temperatures of wood chips could be varied on the basis of total cellulose and hemicellulose content, as well as moisture content of the wood chips [23]. It was reported that smoke generation temperature decreases while the moisture content of wood chips increases [24,25]. Malarut and Vangnai [23] reported that wood chips with low amounts of total cellulose and hemicellulose tended to have a low smoke generation temperature. On the other hand, the ΣPAH4 content in all smoke flavored salmons was lower than the maximum limit, 12 µg/kg [26].

The total content of ΣPAH4 varied between 0.72–3.18 µg/kg in smoked salmon as reported by Zachara et al. [20]. On the other hand, Storelli et al. [19] quantified the total amount of PAH in commercially smoked Danish, Scottish, and Norwegian salmon to be 56.8, 96.2, and 61.6 ng/g (wet weight), respectively.

PAHs in smoked foods are formed by the incomplete combustion and pyrolysis of organic macromolecules. Smoking leads to the formation of smaller volatile molecules, such as phenols, which are responsible for flavor, aroma, color, and conservation [15]. The low PAH content obtained in the present study in salmon fillets smoke flavored with different wood chips compared with others could be attributed to smoking conditions (smoking temperature, smoking time, smoking vehicle, fish origin etc.) [7,19,20]. Indeed, in the present study, salmon fillets were cold smoke flavored at 24˚C for 5 min and stored for 3 h at the same temperature, and a modern smoking gun was used for smoking process. It was reported that a 200% increase in the average sum of 25 PAHs in smoked salmon was associated with hot smoking (65–80˚C) *vs*. cold smoking (15–30˚C) [13].

## Polycyclic aromatic hydrocarbon contents of barbecued salmon fillets

The PAH contents of barbecued salmon fillets are given in Table 1. While BaA and Chry were determined only in non-smoke flavored raw salmon fillets (CI and CII), all PAHs analyzed in the present study were detected in non-smoke flavored barbecued salmon fillets (CI and CII), however, the amounts of some of them were <LOQ. Although, BaP was not detected in all raw salmon fillets, it was determined in all barbecued salmon fillets at the range of 0.25 to 1.04 ng/g. Additionally, we did not detect IncdP in barbecued salmon fillets smoke flavored with cherry and hickory wood chips; DahA as well was not determined in barbecued salmon fillets smoke flavored with bourbon soaked oak, cherry, and hickory wood chips.

BaA was detected in all barbecued salmon fillets at a range of 0.27 and 1.06 ng/g. While the lowest amount of BaA was determined in barbecued salmon fillets smoke flavored with hickory wood chips, the highest concentration was found in barbecued salmon fillets smoke flavored with apple wood chips. As reported earlier, BaA was detected at a concentration rate of 7.82 ng/g in salmon barbecued with wood charcoal and at a rate of 1.82 ng/g in salmon barbecued with coconut charcoal [27].

Herein, Chry was detected in all barbecued salmon fillets and its amount varied between 0.33 and 1.08 ng/g. While the lowest amount of Chry was estimated in barbecued salmon fillets smoke flavored with bourbon soaked oak wood chips, the highest content was determined in barbecued salmon fillets smoke flavored with oak wood chips. As previously reported, Chry was quantified at the level of 20.60 ng/g in salmon barbecued with wood charcoal and 3.57 ng/g was detected in salmon barbecued with coconut charcoal [27].

In this study, BbF was detected in all non-smoke flavored and smoke flavored barbecued salmon fillets. While BbF was not measured in barbecued salmon fillets smoke flavored with bourbon soaked oak and hickory wood chips, its amount varied between 0.41 and 0.82 ng/g in other samples. It has to be noted that Viegas et al. [27] found BbF at a level of 4.87 ng/g in salmon barbecued with wood charcoal and an amount of 1.51 ng/g in salmon barbecued with coconut charcoal [27].

Therein, BkF was detected in all barbecued salmon fillets; however, BkF was not determined in all samples. As stated by others, BkF was quantified at a rate of 1.17 ng/g in salmon barbecued with wood charcoal and at a level of 0.25 ng/g in salmon barbecued with coconut charcoal [27].

In the current study, BaP was measured in all barbecued salmon fillets at a concentration rate of 0.23 and 1.04 ng/g. The maximum limit of BaP was determined to be 5 μg/kg in heat-treated meat products as stipulated by EU Commission Regulation [26]. Herein, the amount of BaP was lower than the aforementioned limit for all barbecued salmon fillets. While the lowest concentration of BaP was found in barbecued salmon fillets smoke flavored with bourbon soaked oak wood chips, the highest level was determined in barbecued salmon fillets smoke flavored with oak wood chips; same as Chry. BaP was quantified at a level of 4.72 ng/g in salmon barbecued with wood charcoal and a level of 1.36 ng/g was detected in salmon barbecued with coconut charcoal as reported by Viegas et al. [27].

In this study, DahA was not estimated in barbecued salmon fillets smoke flavored with bourbon soaked oak, cherry, and hickory wood chips. While DahA was detected in other samples, its amount was not determined in the present study. As recorded by Viegas et al. [27], DahA was measured at a level of 1.22 ng/g in salmon barbecued with wood charcoal and quantified at a level of 0.31 ng/g in salmon barbecued with coconut charcoal.

BghiP was detected in all non-smoke flavored and smoke flavored barbecued salmon fillets. While BghiP was not quantified in non-smoke flavored barbecued salmon fillets; its amount varied between 0.47 and 0.79 ng/g in stored and smoke flavored barbecued salmon with bourbon soaked oak, cherry and hickory wood chips. Notably, the lowest amount of BghiP was found in non-smoke flavored and non-stored barbecued salmon, whereas, the highest level was detected in barbecued salmon fillets smoke flavored with apple wood chips. As reported elsewhere, BghiP was measured at a level of 1.74 ng/g in salmon barbecued with wood charcoal and at a rate of 0.70 ng/g in salmon barbecued with coconut charcoal [27].

Herein, IncdP was not measured in barbecued salmon fillets smoke flavored with cherry and hickory wood chips, while it was detected but not quantified in non-smoke flavored barbecued salmon fillets as well as non-stored and smoke flavored barbecued salmon fillets with bourbon soaked oak wood chips. While IncdP was determined at a level of 1.18 ng/g in salmon barbecued with wood charcoal, it was not detected in salmon barbecued with coconut charcoal as stated by Viegas et al. [27]. The amount of PAHs reported by Viegas et al. [27] was higher than that recorded in our present; the finding which might be attributed to the differences in cooking conditions.

While light PAHs were determined in raw salmon fillets, barbecuing would cause the formation of both light and heavy PAHs. Therefore, barbecuing would increase the contents of ΣPAH4 and ΣPAH8 in all non-smoke flavored salmon fillets and fillets smoke flavored with

different wood chips. ΣPAH4 and ΣPAH8 of barbecued salmon fillets varied between 0.84 and 3.90 ng/g and 0.84 and 5.18 ng/g, respectively. The ΣPAH4 contents in all barbecued salmon fillets were lower than the legal limit set for heat-treated meat products.

While the lowest ΣPAH4 and ΣPAH8 were statistically determined in barbecued salmon fillets smoke flavored with bourbon soaked oak and hickory wood chips, the highest concentration of ΣPAH4 and ΣPAH8 were statistically detected in barbecued salmon fillets smoke flavored with apple wood chips. Smoke flavoring with bourbon soaked oak, cherry, and hickory wood chips statistically decreased the levels of ΣPAH4 and ΣPAH8 compared to non-smoke flavored samples. Similarly, Janoszka [28] proposed that antioxidants, such as polyphenols, could prevent the oxidation and polymerization of hydrocarbons produced from decomposition of fatty acids and protein and lead to lower level of PAHs compared to control group without antioxidant. On the other hand, in the present study, smoke flavoring with oak and apple wood chips significantly increased the levels of ΣPAH4 and ΣPAH8. The inhibitory or promoter effects of smoke flavoring process on PAH content could be attributed to the phenolic compounds and smoke generation temperatures of wood chips [13,15].

### Content of heterocyclic aromatic amine in barbecued salmon fillets

The LOD, LOQ, and recovery rates of heterocyclic aromatic amine (HAAs) are also presented in Fig 1. The obtained values were comparable with those reported in the literature for HAAs [17,29,30].

HAAs contents of non-smoke flavored barbecued salmon fish fillets and smoke flavored one with different wood chips are given in Table 2. IQx (0.04 ng/g), IQ (up to 0.10 ng/g), MeIQx (up to 0.09 ng/g), MeIQ (up to 0.08 ng/g), 7,8-DiMeIQx (< LOQ values), 4,8-DiMeIQx (up to 0.13 ng/g), PhIP (up to 0.89 ng/g), AαC (up to 0.04 ng/g) and MeAαC (< LOQ values) were detected at various levels in barbecued salmon fillets.

In the present study, IQx was not detected in barbecued salmon fillets smoke flavored with bourbon soaked oak wood chips, however, it was detected but not quantified in barbecued salmon fillets smoke flavored with apple and cherry wood chips. The highest amount of IQx was recorded in non-smoke flavored barbecued salmon fillets. As reported by others, IQx was

**Table 2. Heterocyclic aromatic amine (HAA) contents of the barbecued salmon fillets non-smoke flavored and smoke flavored with different smoking wood chips (ng/g).**

| Group | n | IQx | IQ | MeIQx | MeIQ | 7,8-DiMeIQx | 4,8-DiMeIQx | PhIP | AαC | MeAαC | ΣHAAs |
|-------|---|-----|-----|-------|------|-------------|-------------|------|-----|-------|-------|
| C-I | 4 | 0.04±0.02 | nd | 0.09±0.02 | nq | nq | 0.03±0.01 | 0.89±0.09 | 0.04±0.02 | nq | 1.09±0.11 a |
| C-II | 4 | 0.03±0.01 | 0.04±0.02 | nq | nq | nq | nq | 0.77±0.07 | nq | nq | 0.84±0.10 b |
| S-I | 4 | 0.03±0.01 | 0.07±0.03 | nd | 0.08±0.02 | nd | 0.13±0.03 | 0.36±0.05 | 0.04±0.02 | nd | 0.71±0.07 bcd |
| S-II | 4 | nq | 0.23±0.04 | nd | nd | nd | 0.05±0.02 | 0.32±0.04 | nq | nd | 0.60±0.06 cd |
| S-III | 4 | nd | nq | nq | 0.07±0.02 | nq | nq | 0.48±0.06 | nd | nq | 0.55±0.07 d |
| S-IV | 4 | nq | 0.10±0.02 | nd | nq | nq | nq | 0.09±0.02 | nd | nd | 0.19±0.03 e |
| S-V | 4 | 0.02±0.01 | 0.04±0.02 | 0.09±0.02 | 0.07±0.02 | nd | nq | 0.58±0.09 | nd | nd | 0.80±0.11 bc |
| Sign | | | | | | | | | | | ** |

n: number of salmon fillet; C-I: control non-smoke flavored, non-stored salmon fillets; C-II: control non-smoke flavored salmon fillets stored at 24˚C for 3 h; S-I: salmon fillets smoke flavored over oak wood chips and stored at 24˚C for 3 h; S-II: salmon fillets smoke flavored over apple wood chips and stored at 24˚C for 3 h; S-III: salmon fillets smoke flavored over bourbon soaked oak wood chips and stored at 24˚C for 3 h; S-IV: salmon fillets smoke flavored over cherry wood chips and stored at 24˚C for 3 h; S-V: salmon fillets smoke flavored over hickory wood chips and stored at 24˚C for 3 h; nd: not detected (<LOD); nq: not quantified (LOD<...<LOQ); Sign: significance

**: $P < 0.01$; a-e: means with different letters in the same column are significantly different ($P < 0.05$).

determined at a rate of 0.38 ng/g in oven cooked salmon [31], and up to 0.22 ng/g in different types of fish (salmon, mackerel, sardine, whiting, trout and sea bass) cooked with different cooking methods (microwave, pan-frying, oven, hot plate and barbecuing) [32].

Herein, IQ could was not detected in non-smoke flavored barbecued salmon fillets; however, it was detected but not quantified in barbecued salmon fillets smoke flavored with bourbon soaked oak wood chips. The highest concentration was found in barbecued salmon fillets smoke flavored with apple wood chips. IQx was not measured in cooked salmon, sardine, sea trout, rainbow trout, and codfish as reported by others [1,27,29]. On the other hand, IQ was determined at the level of 0.71 ng/g in various types of fish cooked with different cooking methods as stated by Oz and Kotan [32].

MeIQx is one of the most abundant HAAs in cooked meat and fish [33]. In this study, MeIQx was not detected in barbecued salmon fillets smoke flavored with oak, apple, and cherry wood chips, however, it was detected but not quantified in non-smoke flavored barbecued salmon fillets stored at 24˚C for 3 h and smoke flavored fillets with bourbon soaked oak wood chips. The highest concentration (0.09 ng/g) was recorded in non-smoke flavored, non-stored barbecued salmon fillets and barbecued salmon fillets smoke flavored with hickory wood chips. MeIQx was quantified up to the level of 5 ng/g in pan-fried salmon as stated by Gross and Grüter [34], and estimated up to the concentration of 0.87 ng/g in barbecued and pan-fried salmon and sardine as reported by Iwasaki et al. [35].

Therein, MeIQ was not detected in barbecued salmon fillets smoke flavored with apple wood chips, however, it was detected but not quantified in non-smoke flavored, non-stored barbecued salmon fillets, non-smoke flavored barbecued salmon fillets stored at 24˚C for 3 h, and barbecued salmon fillets smoke flavored with cherry wood chips. The highest level was found in barbecued salmon fillets smoke flavored with oak wood chips. As reported by others, MeIQ was not detected in cooked sea trout, codfish, and rainbow trout [29,36].

In the present study, 7,8-DiMeIQx was not detected in barbecued salmon fillets smoke flavored with oak, apple, and hickory wood chips, however, it was detected but not quantified in other salmon fillets. On the other hand, 7,8-DiMeIQx was determined up to the level of 0.06 ng/g in different types of fish cooked with different cooking methods as recorded by Oz and Kotan [32].

Although, 4,8-DiMeIQx was detected in all barbecued samples, it was not determined in non-smoke flavored barbecued salmon fillets stored at 24˚C for 3 h and smoke flavored one with bourbon soaked oak, cherry, and hickory wood chips. The highest amount was recorded in barbecued salmon fillets smoke flavored with oak wood chips. Obviously, 4,8-DiMeIQx was determined at a rate of 1.93 ng/g in pan-fried salmon and a level of 1.66 ng/g in salmon cooked with oven as stated by Puangsombat et al. [31].

In the literature, PhIP has been described as one of the most common HAAs in meats undergo heat treatment [29,35]. PhIP was determined in all non-smoke flavored barbecued salmon fillets and smoke flavored one with different wood chips study at a level ranged between 0.09 and 0.89 ng/g. The highest concentration was found in non-smoke flavored and non-stored barbecued salmon. PhIP was determined up to the level of 28.9 ng/g in grilled and barbecued salmon as investigated by Costa et al. [1].

AαC was not estimated in barbecued salmon fillets smoke flavored with bourbon soaked oak, cherry, and hickory wood chips, however, it was detected but not quantified in non-smoke flavored barbecued salmon fillets stored at 24˚C for 3 h. The highest concentration (0.04 ng/g) was found in non-smoke flavored, non-stored barbecued salmon fillets and barbecued salmon fillets smoke flavored with oak wood chips. Notably, AαC was not detected in cooked salmon, mackerel, sardine, sea bass cod fish, and rainbow trout as reported elsewhere [32,36]. On the other hand, it was determined up to the level of 0.62 ng/g in barbecued salmon as demonstrated by Viegas et al. [27].

MeAαC was not detected in barbecued salmon fillets smoke flavored with oak, apple, cherry, and hickory wood chips; however, it was detected but not quantified in other salmon fillets. In line with our finding, MeAαC was not detected in cooked salmon, mackerel, sardine, whiting, rainbow trout, sea bass, and codfish as reported elsewhere [29,32,36]. At variance, Viegas et al. [27] have found MeAαC at a concentration level of 0.50 ng/g in barbecued salmon.

Barbecuing generally increases the levels of HAA [27,36]. Oz and Kotan [32] reported the highest trend of HAA in barbecued fishes as following: salmon (5.72 ng/g) > sea bass (3.90 ng/g) > rainbow trout (3.44 ng/g) > sardine (2.74 ng/g) > mackerel (1.77 ng/g) > whiting (0.97 ng/g). On the other hand, the total content of HAA (MeIQx, 4,8-DiMeIQx and PhIP) in barbecued salmon ranged between 0.16 and 23.84 ng/g as recorded by Iwasaki et al. [35].

Herein, we found that barbecuing process had a substantial effect on the total HAA content that are ranged from 0.19 to 1.09 ng/g. We also found that smoke flavoring with different wood chips may cause a reduction in the total HAAs. The occurrence of HAAs in foods depend on many factors, such as cooking methods, cooking temperatures, cooking time, cooking equipment, presence of precursors, enhancers and inhibitors, lipid contents, antioxidants, and food composition [33]. Radical reactions had an important role on the formation of HAAs, and it is, therefore, expected that antioxidants can reduce the formation of HAAs in meat and meat products [37]. It was reported that antioxidants could inhibit the formation of HAAs because it interfere with different stages of HAAs formation [38,39]. Indeed, antioxidants may function as free radical scavengers and they may act in the early stages of the Maillard reaction prior to the Amadori rearrangement [40]. Weisburger et al. [41] reported that the tea polyphenols could act as competitive traps for the Maillard reaction intermediates leading to HAA formation. Britt et al. [40] reported that the formation of HAAs during frying of patties with cherry tissue was inhibited by components in the cherry tissue. Therefore, the inhibitory effect of smoke flavoring on the formation of HAAs in salmon fillets could be associated with antioxidant effect of smoking compounds.

The lowest HAA content was statistically recorded in barbecued salmon fillets smoke flavored with cherry wood chips (0.19 ng/g), followed by barbecued salmon fillets smoke flavored with bourbon soaked oak wood chips (0.55 ng/g) ≤ in barbecued salmon fillets smoke flavored with apple wood chips (0.60 ng/g) ≤ in barbecued salmon fillets smoke flavored with oak wood chips (0.71 ng/g) ≤ in barbecued salmon fillets smoke flavored with hickory wood chips (0.80 ng/g) ≤ in non-smoke flavored barbecued salmon fillets stored at 24°C for 3 h (0.84 ng/g) < in non-smoke flavored, non-stored barbecued salmon fillets. These results indicate that smoke flavoring with all types of wood chips prior barbecuing would decrease the total HAA contents in barbecued salmon fillets, the reduction which is statistically significant in barbecued salmon fillets smoke flavored with apple, bourbon soaked oak, and cherry wood chips compared to non-smoke flavored samples. The reduction in samples smoke flavored with apple, bourbon soaked oak, and cherry wood chips may be related to high lignin content of these wood chips. Phenolic compounds that act as antioxidative compounds in wood smoke are formed by pyrolysis of lignin [42]. Pöhlmann et al. [22] reported that lignin found in the wood chips at different levels, plays an important role in antioxidative activity of smoked products. Britt et al. [40] found that the concentrations of total HAAs and PhIP (which is the principal HAA in cooked meat products) were lower in fried patties with cherry tissue than other patties without cherry tissue.

It is declared that the intake of HAAs per person was 60–1820 ng/day, while the maximum intake level of HAAs per person was estimated to be 5000 ng [43]. It is believed that the differences in eating habits and analytical methods would substantially affect the intake levels stated above. On the other hand, Skog [44] estimated the daily intake of HAAs per person between 0

and 15 μg. In the present study, the total intake of HAAs was less than 1 μg even in the case of consuming 100 g of barbecued salmon fillets that contain the highest total HAAs content.

## Conclusions

The present study indicates that BaA and Chry are often present in raw salmon fillets and smoke flavoring process causes an increase in the total content of PAH. BaP and the total PAH contents in salmon fillets are not exceeding the legal limits. Smoke flavoring wood chips showed not only inhibitory but also promoting affects on PAHs contents of barbecued salmon fillets. In this regard, bourbon soaked oak, cherry, and hickory wood chips are the most important wood chips types that could reduce the content of PAHs in barbecued salmon fillets. Barbecuing process did not cause to occur at high levels of HAAs. Interestingly, the total HAA contents of barbecued salmon fillets smoke flavored with apple, bourbon soaked oak, and cherry wood chips were lower than that of non-smoke flavored barbecued samples. In the light of the results of the present study, it can be said that the most important smoke flavoring wood chips types are bourbon soaked oak and cherry smoking wood chips due to the inhibitory effects of them on the PAH and HAA contents.

## Acknowledgments

The author is grateful to Prof. Fatih Oz for his generous support regarding the analysis of heterocyclic aromatic amines.

## Author Contributions

**Conceptualization:** Emel Oz.

**Formal analysis:** Emel Oz.

**Investigation:** Emel Oz.

**Methodology:** Emel Oz.

**Validation:** Emel Oz.

**Writing – original draft:** Emel Oz.

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
