## [Decision Letter · Decision Letter 0]

28 Oct 2019

PONE-D-19-23526

The effects of smoking with different wood chips and barbecuing on the formation of polycyclic aromatic hydrocarbons and heterocyclic aromatic amines in salmon

PLOS ONE

Dear Dr. Oz,

Thank you for submitting your manuscript to PLOS ONE. After careful consideration, we feel that it has merit but does not fully meet PLOS ONE’s publication criteria as it currently stands. Therefore, we invite you to submit a revised version of the manuscript that addresses the points raised during the review process.

We would appreciate receiving your revised manuscript by Dec 12 2019 11:59PM. To enhance the reproducibility of your results, we recommend that if applicable you deposit your laboratory protocols in protocols.io, where a protocol can be assigned its own identifier (DOI) such that it can be cited independently in the future. For instructions see: http://journals.plos.org/plosone/s/submission-guidelines#loc-laboratory-protocols

We look forward to receiving your revised manuscript.

Kind regards,

Chon-Lin Lee, Ph.D.

Academic Editor

PLOS ONE

Journal Requirements:

Please provide an amended Funding Statement that declares *all* the funding or sources of support received during this specific study (whether external or internal to your organization) as detailed online in our guide for authors at http://journals.plos.org/plosone/s/submit-now.  Please state what role the funders took in the study.  If any authors received a salary from any of your funders, please state which authors and which funder. If the funders had no role, please state: "The funders had no role in study design, data collection and analysis, decision to publish, or preparation of the manuscript."

Reviewers' comments:

Reviewer's Responses to Questions

**Comments to the Author**

1. Is the manuscript technically sound, and do the data support the conclusions?

Reviewer #1: Partly

Reviewer #2: Partly

2. Has the statistical analysis been performed appropriately and rigorously? 

Reviewer #1: No

Reviewer #2: No

3. Have the authors made all data underlying the findings in their manuscript fully available?

Reviewer #1: Yes

Reviewer #2: Yes

4. Is the manuscript presented in an intelligible fashion and written in standard English?

Reviewer #1: Yes

Reviewer #2: Yes

5. Review Comments to the Author

Reviewer #1: General opinion

The study presents the results of original research dealing with the impact of smoking and barbecuing on the formation of polycyclic aromatic hydrocarbons and heterocyclic aromatic amines in farmed salmon fillets. To my knowledge and conducted search the results of this study have not been published elsewhere. In my opinion, the paper should not be published in this form for several reasons.

My biggest concerns about this paper are following:

- the experiment was performed using two fillets of the same fish, which means 1 fish per treatment which is, in my opinion, too low number to draw conclusions from.

- The smoking gun, a device used in the home kitchen and catering services to infuse smoke flavor to foods and drinks, was used as an alternative to smoking process and in duration of only 5 minutes. This is not smoking, just flavoring therefore I just do not see the practical application of these results, especially it cannot be concluded that a smoking process affected the PAHs content.

- Experiment was not described or justified in sufficient detail: why did you apply the smoke for 5 min at 24 °C?; what is purpose of storage 3 h storage at 24…etc.

- Described statistics were not applied for all data (standard deviation and variance analysis for BaA and Chry in table 2 missing; in table 3 analysis of variance was done only for sum of HAAs).

- Sums of PAH4 and 8 are not described, abbreviations are not explained (lines 31, 34 etc.)

- some English editing is necessary throughout the text

Specific comments:

Lines31; 34 give full names before abbreviations

Lines 43-45 Fish has an important potential to meet the growing nutrient needs due to its high protein, omega-3 polyunsaturated fatty acids (PUFAs), especially EPA…

Lines 49 …due to the content of PUFAs.

Line 104. Salmon fillets were separated to seven groups and two groups were selected as the control.

Lines 115-118: Authors kempt the salmon fillets non-smoked for 3 h at 24°C to determining the effect of the storage on the quality of the salmon fillets. What quality?

Line 125: kitchen blender grinds. It does not homogenize

Lines 162-163: in two replicas per analysis or ? Please clarify this

Line 240. You analyzed the farmed fish, so its diet may be the why PAHs are present in the raw material as well.

Line 248 PAHS in smoked foods could be…

Line 255. What literature are you referring to?

Reviewer #2: General comment:

The topic of this article is of interest because there is a general lack of knowledge concerning the influence of wood type on the formation of PAHs. However, some conclusions are not new (for example, BaA and Chry are often present in raw salmon fillets and smoking process caused an increase in total PAH contents), and controversial results were obtained, thus discussion of the results needs improvement to justify the obtained results (why bourbon soaked oak, cherry and hickory smoking wood chips reduce content of PAHs in barbecued? total HAA contents of barbecued salmon fillets smoked with apple, bourbon soaked oak and cherry wood chips were lower than those of non smoked barbecued samples salmon fillets.

Detailed comments:

In the abstract and at the end of introduction section authors describe that the aim of this paper is to understand the effect of smoking with five smoking wood chips and barbecuing on the levels of PAHs and HAAs in salmon fillets, however, the experimental planning describes the use of a modern smoking gun for the smoking process. This means that it is not the industrial smoking but the use of a novel equipment for smoking at home? It should be clarified.

Abstract needs to be rewritten to include some data and not only general description.

The reproducibility of the method is not mentioned, only LOD, LOQ and recoveries.

Concerning fish sampling (n=14) how can authors be sure that fishes have all the same contamination?

Table 2. Polycyclic aromatic hydrocarbon (PAH) contents of the raw and barbecued salmon fillets non-smoked and smoked with different smoking wood chips (ng/g)

Non-smoked

C-I: control group salmon fillets non-smoked, non-stored;

C-II: control group salmon fillets non-smoked, stored at 24ºC for 3 h;

There is no reason to increase the PAHs content in C-II, except if it losses water and concentrates the initial PAHs content. Please justify.

smoked

S-I: salmon fillets smoked with oak smoking wood chips, stored at 24ºCfor 3 h;

S-II: salmon fillets smoked with apple smoking wood chips, stored at 24ºC for 3 h;

S-I and S-II samples present higher content of PAHs when compared with SII, S-IV and S-V, it is not clear why?

Tables 2 and 3, please include the number of samples analysed

Authors describe that the smoking with all of the smoke wood chips before the barbecuing decreased the total HAA contents of the barbecued salmon fillets and the decrease was statistically significant in the barbecued salmon fillets smoked with apple, bourbon soaked oak and cherry smoke wood chips compared to the samples not smoked. This effect of the smoking process on the formation of HAAs in the salmon fillets could be due to the antioxidant effect of the smoke compounds. More scientific explanations or references are needed concerning this point.

Why the statistical treatment was done only in the sum of PAHs or HAAs and not in the individual compounds?

Abbreviations in the text and in tables should be uniform

IncdP in the text appears as Incdp or as IncdP

Bghip usually it is BghiP

6. PLOS authors have the option to publish the peer review history of their article (what does this mean?). If published, this will include your full peer review and any attached files.

Reviewer #1: No

Reviewer #2: No

---

## [Author Response · Author response to Decision Letter 0]

6 Nov 2019

Plos One

PONE-D-19-23526

“The effects of smoking with different wood chips and barbecuing on the formation of polycyclic aromatic hydrocarbons and heterocyclic aromatic amines in salmon”

Dear Editor and Reviewers,

Thanks a bunch for your criticisms on our MS entitled “The effects of smoking with different wood chips and barbecuing on the formation of polycyclic aromatic hydrocarbons and heterocyclic aromatic amines in salmon”. We have taken all the comments into consideration. All amendments and corrections are highlighted yellow on the revised form of the MS. Herein with the answer point by point as listed below; 

Response to the reviewers:

Review Comments to the Author

Reviewer #1:

General opinion

The study presents the results of original research dealing with the impact of smoking and barbecuing on the formation of polycyclic aromatic hydrocarbons and heterocyclic aromatic amines in farmed salmon fillets. To my knowledge and conducted search the results of this study have not been published elsewhere. In my opinion, the paper should not be published in this form for several reasons.

My biggest concerns about this paper are following:

The experiment was performed using two fillets of the same fish, which means 1 fish per treatment which is, in my opinion, too low number to draw conclusions from.

Thank you. Please note that a total of 28 salmon fillets obtained from 14 salmon fishes were used as material. Apologize for misunderstanding. This part of has been amended in the text. Please see line 112.

The smoking gun, a device used in the home kitchen and catering services to infuse smoke flavor to foods and drinks, was used as an alternative to smoking process and in duration of only 5 minutes. This is not smoking, just flavoring therefore I just do not see the practical application of these results, especially it cannot be concluded that a smoking process affected the PAHs content.

Thanks for your comment. You are right, therefore, we used homemade smoking instead of smoking throughout the article.

Experiment was not described or justified in sufficient detail: why did you apply the smoke for 5 min at 24 °C?; what is purpose of storage 3 h storage at 24…etc.

Thanks for this good comment. The purpose of storage for 3 h is to accumulate smoke. Please see line 120-121.

Described statistics were not applied for all data (standard deviation and variance analysis for BaA and Chry in table 2 missing; in table 3 analysis of variance was done only for sum of HAAs).

Standard deviations of individual PAHs in Table 2 have been added to Table. Please see the Table 2. Variance analysis was only performed on ∑PAH4, ∑PAH8 and total HAAs, rather than individual analytes, because it is an important item while considering the toxicity of the compound.

Sums of PAH4 and 8 are not described, abbreviations are not explained (lines 31, 34 etc.)

Thank you. The descriptions have been amended in the abstract. Please see the abstract.

Some English editing is necessary throughout the text

Thank you. The MS has been edited by a colleague who is fluent in English for grammar and syntax error. Please see the article.

Specific comments:

Lines 31; 34 give full names before abbreviations

The acronym has been added as suggested. Please see the abstract.

Lines 43-45 Fish has an important potential to meet the growing nutrient needs due to its high protein, omega-3 polyunsaturated fatty acids (PUFAs), especially EPA…

Lines 49 …due to the content of PUFAs.

Thank you. This has been corrected. Please see the lines 57-58.

Line 104. Salmon fillets were separated to seven groups and two groups were selected as the control.

Thank you. This has been corrected. Please see the line 113.

Lines 115-118: Authors kempt the salmon fillets non-smoked for 3 h at 24°C to determining the effect of the storage on the quality of the salmon fillets. What quality?

Apologies. This sentence has been deleted from the text. 

Line 125: kitchen blender grinds. It does not homogenize

“homogenized” has been changed to “minced”. Please see the line 129.

Lines 162-163: in two replicas per analysis or ? Please clarify this

This means we analyzed 2 fishes/4 fillets for each analyze.

Line 240. You analyzed the farmed fish, so its diet may be the why PAHs are present in the raw material as well.

Thank you. This statement has been amended as suggested. Please see the lines 227-228.

Line 248 PAHS in smoked foods could be…

Thank you. This statement has been corrected as suggested. 

Line 255. What literature are you referring to?

Supporting references have been added to the text. Please see the line 260.

Reviewer #2: 

General comment:

The topic of this article is of interest because there is a general lack of knowledge concerning the influence of wood type on the formation of PAHs. However, some conclusions are not new (for example, BaA and Chry are often present in raw salmon fillets and smoking process caused an increase in total PAH contents), and controversial results were obtained, thus discussion of the results needs improvement to justify the obtained results (why bourbon soaked oak, cherry and hickory smoking wood chips reduce content of PAHs in barbecued? total HAA contents of barbecued salmon fillets smoked with apple, bourbon soaked oak and cherry wood chips were lower than those of non smoked barbecued samples salmon fillets.

Thanks for your query. The inhibitory effect of some smoking wood chips on the formation of HAAs and PAHs could be due to the content of smoking wood chips. In the literature, it was stated that antioxidants could prevent the oxidation and polymerization of hydrocarbons produced from decomposition of fatty acids and protein and lead to low level of PAHs. Details have been added to text as suggested. Please see the lines 343-350, 437-449, and 460-468.

Detailed comments:

In the abstract and at the end of introduction section authors describe that the aim of this paper is to understand the effect of smoking with five smoking wood chips and barbecuing on the levels of PAHs and HAAs in salmon fillets, however, the experimental planning describes the use of a modern smoking gun for the smoking process. This means that it is not the industrial smoking but the use of a novel equipment for smoking at home? It should be clarified.

Thank you. Therefore, smoking process has been replaced with homemade smoking throughout the article.

Abstract needs to be rewritten to include some data and not only general description.

Thank you. The abstract has been updated with some data as suggested. Please see the abstract

The reproducibility of the method is not mentioned, only LOD, LOQ and recoveries.

We have published some related articles on the determination of HAAs and PAHs in meat and meat products. As it is a lab work, we thought that only aforementioned parameters (LOD, LOQ and recovery) are enough for method performance.

Concerning fish sampling (n=14) how can authors be sure that fishes have all the same contamination?

We are also not sure that all fishes used as material in the present study have the same PAH contamination. Therefore, a randomized design was used in statistical evaluation of the data.

Table 2. Polycyclic aromatic hydrocarbon (PAH) contents of the raw and barbecued salmon fillets non-smoked and smoked with different smoking wood chips (ng/g)

Non-smoked

C-I: control group salmon fillets non-smoked, non-stored;

C-II: control group salmon fillets non-smoked, stored at 24ºC for 3 h;

There is no reason to increase the PAHs content in C-II, except if it losses water and concentrates the initial PAHs content. Please justify.

Absolutely, you are right. However, the water content was not significantly decreased between C-I and C-II (data belongs to our lab). In addition, while the PAH content in C-II was higher than that of C-I, this was not significant as statistical (P > 0.05). 

smoked

S-I: salmon fillets smoked with oak smoking wood chips, stored at 24ºCfor 3 h;

S-II: salmon fillets smoked with apple smoking wood chips, stored at 24ºC for 3 h;

S-I and S-II samples present higher content of PAHs when compared with SII, S-IV and S-V, it is not clear why?

Thank you. Details have been added to text as suggested. Please see the lines 240-248.

Tables 2 and 3, please include the number of samples analysed

Thank you. The numbers (n values) of samples analyzed have been added to Table 2 and Table 3.

Authors describe that the smoking with all of the smoke wood chips before the barbecuing decreased the total HAA contents of the barbecued salmon fillets and the decrease was statistically significant in the barbecued salmon fillets smoked with apple, bourbon soaked oak and cherry smoke wood chips compared to the samples not smoked. This effect of the smoking process on the formation of HAAs in the salmon fillets could be due to the antioxidant effect of the smoke compounds. More scientific explanations or references are needed concerning this point.

Thank you. Detailed discussion has been added to the text. Please see the lines 437-449 and 460-468.

Why the statistical treatment was done only in the sum of PAHs or HAAs and not in the individual compounds?

Thank you for your attention. Due to the fact that the amount of many of the individual PAHs and HAAs were lower than LOD or LOQ (nd or nq), statistical analyses were performed only for the sum of PAHs or HAAs.

Abbreviations in the text and in tables should be uniform

Thank you. This has been amended as suggested. 

IncdP in the text appears as Incdp or as IncdP

Incdp has been changed to IncdP

Bghip usually it is BghiP

Bghip has been changed to BghiP

The revised manuscript has been resubmitted to your journal. I look forward to your positive response as soon as possible. Is there anything I can do for that, please don’t hesitate to contact me.

Sincerely yours,

Dr. Oz

---

## [Decision Letter · Decision Letter 1]

17 Dec 2019

PONE-D-19-23526R1

Effects of homemade smoking with different wood chips and barbecuing on the formation of polycyclic aromatic hydrocarbons and heterocyclic aromatic amines in salmon fillets

PLOS ONE

Dear Dr. Oz,

Thank you for submitting your manuscript to PLOS ONE. After careful consideration, we feel that it has merit but does not fully meet PLOS ONE’s publication criteria as it currently stands. Therefore, we invite you to submit a revised version of the manuscript that addresses the points raised during the review process.

We would appreciate receiving your revised manuscript by Jan 31 2020 11:59PM. To enhance the reproducibility of your results, we recommend that if applicable you deposit your laboratory protocols in protocols.io, where a protocol can be assigned its own identifier (DOI) such that it can be cited independently in the future. For instructions see: http://journals.plos.org/plosone/s/submission-guidelines#loc-laboratory-protocols

We look forward to receiving your revised manuscript.

Kind regards,

Chon-Lin Lee, Ph.D.

Academic Editor

PLOS ONE

Reviewers' comments:

Reviewer's Responses to Questions

**Comments to the Author**

1. If the authors have adequately addressed your comments raised in a previous round of review and you feel that this manuscript is now acceptable for publication, you may indicate that here to bypass the “Comments to the Author” section, enter your conflict of interest statement in the “Confidential to Editor” section, and submit your "Accept" recommendation.

Reviewer #1: All comments have been addressed

2. Is the manuscript technically sound, and do the data support the conclusions?

Reviewer #1: Yes

3. Has the statistical analysis been performed appropriately and rigorously? 

Reviewer #1: No

4. Have the authors made all data underlying the findings in their manuscript fully available?

Reviewer #1: No

5. Is the manuscript presented in an intelligible fashion and written in standard English?

Reviewer #1: Yes

6. Review Comments to the Author

Reviewer #1: Journal: Plos One

Research paper: "The effects of smoking with different wood chips and barbecuing on the formation of polycyclic aromatic hydrocarbons and heterocyclic aromatic amines in salmon" (PONE-D-19-23526_R1)

The author accepted most of the suggested corrections; still one change is necessary before the paper can be accepted for publication. Smoking is a technological process that includes salting, drying and smoke application. Please change homemade smoking to smoke flavoring throughout the text and update the text accordingly.

Title suggestion

Effects of smoke flavoring using different wood chips and barbecuing on the formation of polycyclic aromatic hydrocarbons and heterocyclic aromatic amines in salmon fillets

7. PLOS authors have the option to publish the peer review history of their article (what does this mean?). If published, this will include your full peer review and any attached files.

Reviewer #1: No

---

## [Author Response · Author response to Decision Letter 1]

18 Dec 2019

Plos One

PONE-D-19-23526R1

“Effects of homemade smoking with different wood chips and barbecuing on the formation of polycyclic aromatic hydrocarbons and heterocyclic aromatic amines in salmon fillets”

Dear Editor and Reviewers,

Thanks a bunch for your criticisms on our MS entitled “Effects of homemade smoking with different wood chips and barbecuing on the formation of polycyclic aromatic hydrocarbons and heterocyclic aromatic amines in salmon fillets”. We have taken all the comments into consideration. All amendments and corrections are highlighted yellow on the revised form of the MS. Herein with the answer point by point as listed below; 

Response to the reviewers:

Review Comments to the Author

Reviewer #1:

Research paper: "The effects of smoking with different wood chips and barbecuing on the formation of polycyclic aromatic hydrocarbons and heterocyclic aromatic amines in salmon" (PONE-D-19-23526_R1)

The author accepted most of the suggested corrections; still one change is necessary before the paper can be accepted for publication. Smoking is a technological process that includes salting, drying and smoke application. Please change homemade smoking to smoke flavoring throughout the text and update the text accordingly.

Thank you for your valuable comment. The statement of “homemade smoking” has been changed as “smoke flavoring” throughout the manuscript.

Title suggestion:

Effects of smoke flavoring using different wood chips and barbecuing on the formation of polycyclic aromatic hydrocarbons and heterocyclic aromatic amines in salmon fillets

Thank you for your valuable suggestion. Title of the manuscript has been changed according to your suggestion.

The revised manuscript has been resubmitted to your journal. I look forward to your positive response as soon as possible. Is there anything I can do for that, please don’t hesitate to contact me.

Sincerely yours,

Dr. Emel Oz

---

## [Editor Report · Decision Letter 2]

20 Dec 2019

Effects of smoke flavoring using different wood chips and barbecuing on the formation of polycyclic aromatic hydrocarbons and heterocyclic aromatic amines in salmon fillets

PONE-D-19-23526R2

Dear Dr. Oz,

We are pleased to inform you that your manuscript has been judged scientifically suitable for publication and will be formally accepted for publication once it complies with all outstanding technical requirements.

With kind regards,

Chon-Lin Lee, Ph.D.

Academic Editor

PLOS ONE
---

## [Editor Report · Acceptance letter]

26 Dec 2019

PONE-D-19-23526R2 

Effects of smoke flavoring using different wood chips and barbecuing on the formation of polycyclic aromatic hydrocarbons and heterocyclic aromatic amines in salmon fillets 

Dear Dr. Oz:

I am pleased to inform you that your manuscript has been deemed suitable for publication in PLOS ONE. Congratulations! Your manuscript is now with our production department. 

With kind regards,

on behalf of

Dr. Chon-Lin Lee 

Academic Editor

PLOS ONE